# Prediction of CD44 Structure by Deep Learning-Based Protein Modeling

**DOI:** 10.3390/biom13071047

**Published:** 2023-06-28

**Authors:** Chiara Camponeschi, Benedetta Righino, Davide Pirolli, Alessandro Semeraro, Francesco Ria, Maria Cristina De Rosa

**Affiliations:** 1Institute of Chemical Sciences and Technologies ‘‘Giulio Natta’’ (SCITEC)-CNR, 00168 Rome, Italy; chiara.camponeschi@scitec.cnr.it (C.C.); benedetta.righino@scitec.cnr.it (B.R.); davide.pirolli@cnr.it (D.P.); 2Department of Chemistry and Technology of Drugs, Sapienza University of Rome, 00185 Rome, Italy; alessandro.semeraro@uniroma1.it; 3Department of Translational Medicine and Surgery, Section of General Pathology, Università Cattolica del Sacro Cuore, 00168 Rome, Italy; francesco.ria@unicatt.it; 4Fondazione Policlinico Universitario A. Gemelli IRCCS, 00168 Rome, Italy

**Keywords:** artificial intelligence, molecular dynamics simulations, intrinsically disordered regions, immune response, hyaluronan-binding domain

## Abstract

CD44 is a cell surface glycoprotein transmembrane receptor that is involved in cell–cell and cell–matrix interactions. It crucially associates with several molecules composing the extracellular matrix, the main one of which is hyaluronic acid. It is ubiquitously expressed in various types of cells and is involved in the regulation of important signaling pathways, thus playing a key role in several physiological and pathological processes. Structural information about CD44 is, therefore, fundamental for understanding the mechanism of action of this receptor and developing effective treatments against its aberrant expression and dysregulation frequently associated with pathological conditions. To date, only the structure of the hyaluronan-binding domain (HABD) of CD44 has been experimentally determined. To elucidate the nature of CD44s, the most frequently expressed isoform, we employed the recently developed deep-learning-based tools D-I-TASSER, AlphaFold2, and RoseTTAFold for an initial structural prediction of the full-length receptor, accompanied by molecular dynamics simulations on the most promising model. All three approaches correctly predicted the HABD, with AlphaFold2 outperforming D-I-TASSER and RoseTTAFold in the structural comparison with the crystallographic HABD structure and confidence in predicting the transmembrane helix. Low confidence regions were also predicted, which largely corresponded to the disordered regions of CD44s. These regions allow the receptor to perform its unconventional activity.

## 1. Introduction

CD44 is a cell surface receptor that interacts with multiple ligands in the extracellular matrix (ECM), the main one of which is hyaluronic acid (HA) [1]. CD44 is involved in the regulation of cell–cell and cell–matrix interactions, cell proliferation, adhesion, migration, hematopoiesis, and lymphocyte activation [2]. The human *cd44* gene contains 19 variant exons, 10 of which are present in all variants and can be alternatively spliced to generate 38 transcripts. Some of these transcripts encode proteins of different lengths, known as CD44 isoforms, namely CD44vv (Appendix A). The shortest and most abundant isoform from which all variant-generating exons are spliced off is called CD44 standard (CD44s) (P16070-12, ENST00000263398.11) due to its wide expression in all tissues and individuals [3,4,5] compared to the other CD44vv isoforms. The variations in the ratio between CD44s and CD44vv play an important role in several pathological processes [6,7,8]. CD44v6 has been widely confirmed as a marker of cancer with metastatic potential, and its inhibition is proving to be a therapeutic strategy [9,10,11,12]. Other isoforms have been proposed as candidate biomarkers of inflammatory processes such as allergic asthma [13], systemic lupus erythematosus (SLE) [14,15], and multiple sclerosis (MS) [16,17,18].

CD44s consists of three primary domains: an extracellular domain (ECD, residues 1–268), a transmembrane domain (TMD, residues 269–289), and a cytoplasmic, intracellular domain (ICD, residues 290–361). Five structures of CD44 have so far been characterized and deposited in the Protein Data Bank (www.rcsb.org, last website visit 20 February 2022). Among these five coordinate entries, three are X-ray crystallographic structures (PDB entries: 1UUH [19], 4PZ3, and 4PZ4 [20]) solved at resolutions of 2.20, 1.08, and 1.60 Å, respectively, and two are NMR structures (1POZ [19] and 2I83 [21]). They all correspond to the hyaluronan-binding domain (HABD) of the receptor, spanning residues 20–169 [19], which is shared by all CD44vv and is never affected by alternative splicing [16]. So far, a complete structure for full-length CD44 is not available, thus limiting our understanding of how the (extracellular) spliced regions can affect the functional domains of the receptor and their role in physiology and pathology. A recently published computational study focused on analyzing the membrane-proximal stem region of CD44 for the standard isoform and cancer-associated isoform v6 in rats [22]. Only the longest CD44 isoform (P16070-1, length 742 residues), containing all ten variant exons is present in the AlphaFold2 database.

To address these limitations, here, we report the construction of a three-dimensional model of the full-length human CD44s using three different modeling approaches: AlphaFold2 [23], RoseTTAFold [24], and D-I-TASSER [25].

AlphaFold2 uses machine learning to produce computational protein structures at near-experimental-scale resolution [24,26,27], as demonstrated by its outstanding performance at the CASP14 competition (https://predictioncenter.org/, accessed on 23 January 2023). RosettaFold draws inspiration from AlphaFold and its collection of deep learning models. It generates highly accurate predictions of protein structures, uses fewer computing resources than AlphaFold, and enables the accurate modeling of the protein–protein complex. The D-I-TASSER pipeline is an extension of I-TASSER, a server that employs ten threading algorithms to predict the tertiary structure of a protein, relying on threading, ab initio modeling, and replica-exchange Monte Carlo dynamics simulations for atomic-level refinement [28,29,30,31]. The models of CD44s’ structure, which we previously generated in I-TASSER, were of low reliability based on their confidence scores. D-I-TASSER, which integrates deep-convolutional-neural-network-based distance and hydrogen-bonding network predictions to assemble template fragments into a full-length model via replica-exchange Monte Carlo simulations, was then used for comparison.

We then carefully evaluated the quality of the generated models and assessed their stability with the support of molecular dynamics simulations. The extension of the analysis to two CD44 variants of particular interest (CD44v3-10 and CD44v7-10) was also discussed.

## 2. Materials and Methods

### 2.1. Molecular Modeling

The sequence of the standard isoform of CD44 (CD44s), corresponding to the 9 exon-composed transcripts (accession number ENST00000263398.11 in the Ensembl database), was retrieved in FASTA format from the UniProt database [32] (entry code P16070-12). CD44s’ 361-amino-acid-long sequences were modeled using three different algorithms for three-dimensional structure predictions, namely AlphaFold2 [23], RoseTTAFold [24], and D-I-TASSER [25]. Model building was carried out using the relevant online resources with their default settings. AlphaFold2, an artificial intelligence (AI) deep learning system developed by Google’s DeepMind, was used in collaborative mode thanks to the public online notebook (https://colab.research.google.com/github/sokrypton/ColabFold/blob/main/AlphaFold2.ipynb, accessed on 14 December 2022) provided by Google [33]. The methodology of AlphaFold has inspired RoseTTAFold (available at https://robetta.bakerlab.org, accessed on 1 December 2022), which uses transformer-based transfer learning to improve prediction accuracy and, in comparison, is potentially faster while maintaining good accuracy [34]. The Distance-guided Iterative Threading ASSEmbly Refinement (D-I-TASSER) server (available at https://zhanggroup.org//D-I-TASSER/, accessed on 2 May 2023) is based on multiple deep neural network models that generate inter-amino acid residue interactions of contact maps, distance maps, and hydrogen-bond networks [25]. Analysis of the stereochemical quality of the predicted three-dimensional structures was performed with PROCHECK [35] and ProSA-Web [36]. The presence of intrinsically disordered regions in the protein was investigated using the disorder predictor PONDR [37]. AlphaFold2 was also used to perform the prediction and analysis of the CD44v3-10 and CD44v7-10 isoforms.

### 2.2. Molecular Dynamics Simulations

Atomistic molecular dynamics (MD) simulation was employed to confirm the stability of the best model of CD44s. The MD simulation was carried out with the OPLS4 forcefield [38] in the explicit SPC water model using the Desmond-6.8 module, as implemented in the Schrödinger software package (Schrödinger Release 2022-3: Desmond Molecular Dynamics System, D. E. Shaw Research, New York, NY, USA, 2022). The POPC (1-palmitoyl-2-oleoyl-sn-glycero-3-phosphocholine) membrane was properly placed by defining atoms 269–289 as the transmembrane region in the system builder tool of Desmond. As the disordered regions 1–19 and 219–237 penetrated the lipid bilayer, structural models of the extracellular, transmembrane, and cytoplasmic regions were separately modeled using AlphaFold2 and joined according to the membrane topology (Appendix A). Geometry optimization via energy minimization terminated the protein preparation. The system builder tool was used to embed the prepared protein in an orthorhombic box, with each side at a minimum distance of 10 Å from the edge of the protein. The box was filled with the SPC water model, and the system was then neutralized by the addition of Na+ ions. A concentration of 0.15M NaCl salt was also added (see Table 1).

Each system was simulated for 500 ns at a temperature of 300 K and a pressure of 1 atm, saving coordinates and energies every 500 ps. Each simulation was repeated in triplicate with different random seeds for the assignment of the initial velocities. An analysis of trajectories was carried out with Maestro (Schrödinger Release 2022-3: Desmond Molecular Dynamics System, D. E. Shaw Research, New York, NY, USA, 2022. Maestro-Desmond Interoperability Tools, Schrödinger, New York, NY, USA, 2022) and VMD v1.9.3 [19]. As an experimental reference, the crystal structure of the hyaluronan-binding domain of CD44 (pdb entry: 1UUH) also was subjected to MD simulation in explicit solvent (Table 1). Visual inspection was performed using both Maestro and Discovery Studio 2022 (Dassault Systèmes BIOVIA, Discovery Studio, 2022, San Diego: Dassault Systèmes, 2021). All the calculations were carried out on a workstation running the CentOS 7 Linux operating system and equipped with NVIDIA RTX series GPUs based on Ampere architecture.

## 3. Results

### 3.1. CD44s Structure Prediction and Validation

We examined the protein structure prediction of CD44s through three different techniques: D-I-TASSER, AlphaFold2, and RoseTTAFold. The top five models were generated via D-I-TASSER, the new version of I-TASSER which incorporates deep-learning-based spatial restraints. In D-I-TASSER, the accuracy of the models is represented by the estimated TM score (eTM score, ranging between 0 and 1, with higher values indicative of higher model confidence), calculated based on threading template alignments, the contact map satisfaction rate, the mean absolute error between model distance and distance of attention potential, and the simulation convergence of simulations [25] The five models generated via D-I-TASSER for CD44s showed an eTM score of 0.37 to 0.42, below the generally accepted reliability threshold of 0.5 [39]. Of the top 10 templates used by D-I-TASSER for threading, the hyaluronan-binding domain of CD44 ranked first, second, third, fourth, and seventh (Appendix A).

The quality of the top ranking model was evaluated considering the overall stereochemistry via PROCHECK, and the resulting Ramachandran plot showed that 60.5% of the residues were found in the most favorable regions, 31.5% in the additional favorable region, 5.1% in the generously allowed region, and the remaining 2.9% in the disallowed region (Appendix A). The ProSA-Web analysis of the model, applied to test the energy criteria against the average force potential derived from a large set of known protein structures, revealed a Z-score value of 6.67, in the range of the experimentally resolved structures of the same size. Almost all the residues had negative values of interaction energy, whilst only a few residues at the N-terminal of CD44 displayed positive values corresponding to problematic or erroneous parts of a model (Appendix A). The Cα-RMSD between the predicted HABD and the crystallographic 1UUH was 1.3 Å, and the superimposition is shown in Figure 1.

From the input sequence, AlphaFold2 produced an MSA (multiple sequence alignment) which showed that the region spanning residues 20–169, corresponding to HABD, was highly conserved (Figure 2A). Five models were then generated, and their quality is shown in Figure 2B,C. Confidence in AlphaFold2 predictions is expressed through the predicted local distance difference test score (pLDDT) on a scale from 0 to 100, with high values showing higher confidence [40], and through the predicted aligned error (PAE), a per-residue pair distance score indicative of the reliability of pairwise relative positions of amino acids, with low values indicating lower errors. Using the LDDT score, residues 20 to 169 achieved very high confidence (pLDDT > 90). Sequence 269–289, corresponding to the transmembrane alpha helix, was also predicted with good confidence. Two low-confidence regions, from residue 170 to residue 268 and from residue 290 to residue 361, were identified (Figure 2B). The PAE output graph showed low errors in the distances of residues within the regions encompassing residues 20–169 and 270–290 (Figure 2C). Figure 2 shows that residues with a blue color in the PAE graph mirrored the peaks in the pLDDT score. Of note, the most accurate prediction corresponded to the HABD domain of CD44s. The first ranked model generated by AlphaFold2 was chosen and subjected to the structural validation.

The Ramachandran plot generated via PROCHECK showed that 66.9% of residues fell in the most favored region, 23.8% in the additional favorable region, 3.2% in the generously allowed regions, and 6.1% in the disallowed regions. The allowed residues were all located in the low confidence regions 169–268 and 290–361 of the CD44s, with none of them belonging to the HABD (Appendix A). ProSA-Web analysis revealed a Z-score of −4.41, within the range of the experimentally resolved structures of the same size. The ProSA-Web energy profile plot, where positive values mean erroneous regions of the model, indicated that most of the residues had negative scores (Appendix A). The predicted AlphaFold2 structure of CD44s is shown in Figure 2D, and the high confidence prediction for HABD is highlighted by its superposition with the crystallographic 1UUH structure and the relative Cα-RMSD (0.8 Å). Next, we generated structural models of CD44s using RoseTTAFold, and the confidence scores of the five models, based on the global distance test (GDT) function [41], were all equal to 0.48 on a scale from 0.0 (bad) to 1.0 (good). The first-ranked model was chosen for comparison with the best model produced using AlphaFold2, and its quality was evaluated. The per-residue error estimate (Figure 3) indicated high-confidence regions from residue 23 to residue 169 and low-confidence prediction for regions spanning residues 190 to 361.

The stereochemistry of the model was quite good, with a total of 98.1% of the residues in the most favorable regions and allowed regions (87.5% and 10.6%, respectively), 1% in the generously allowed region and only three residues (1%) in disallowed regions, corresponding to residues Thr263, Ile143, and Asn39 (Appendix A). ProSA-Web analysis showed a Z-score of −5.51, within the range characteristic for native proteins, indicating the good quality of the built model, and, as for AlphaFold2, the residue energies of the model were largely negative (Appendix A). The predicted RoseTTAFold model was then aligned with the experimentally solved structure of the HABD, and the calculated Cα-RMSD was 1.3 Å (Figure 3B).

To further investigate the results from the deep learning approaches and to correlate low-confidence regions with intrinsic disorder in proteins, we also subjected CD44 sequences to the bioinformatic tool PONDR to identify intrinsically disordered regions (IDRs) in the receptor. CD44s were predicted to contain a large segment of IDRs that span residues 290–349. Few scores above 0.5, indicative of disordered residues, were observed for few residues, including residues 127–136, which belong to a loop region between β7 and β8 strands (Appendix A). Low PONDR scores were observed for the 269–289 region, which corresponds to the transmembrane helix, as predicted via AlphaFold2 with high confidence (Figure 2B,D) and with low confidence via RoseTTAFold (Figure 3A,B). AI-based folding predictions, more for AlphaFold2 than D-I-TASSER and RoseTTAFold, showed a predominantly non-structurally organized long C-terminal which agreed with the analysis of intrinsic disorder for this domain. On the whole, the lack of tertiary structure or the existence of conformational ensembles for this domain correlated well with the low-confidence prediction regions of AlphaFold2. At the same time, the calculated RMSD values between the crystallographic and the modeled HABD (0.8 Å, 1.3 Å, and 1.3 Å for AlphaFold2, D-I-TASSER, and RoseTTAFold, respectively) indicated the major ability of AlphaFold2 to accurately predict the conformation of the hyaluronan-binding domain. This was also confirmed by the excellent PROCHECK results and TM and Molprobity score values (Table 2), which show that the quality of the AlphaFold2 model was comparable to that of the crystal structure.

### 3.2. Stability Analysis by MD Simulations

To investigate the stability of the model predicted using AlphaFold2, the structural dynamics of CD44s, inserted in a lipid bilayer and solvated in water and sodium chloride, were determined using MD simulations (Figure 4A). To better evaluate the conformational landscape of the structured region spanning residues 20–169, MD simulation of the crystallographic HABD (pdb entry 1UUH) in water and sodium chloride was also run. Three different MD replicates were assessed, and the stability of each model was assessed by monitoring, with respect to the starting structure, the RMSD, the root mean square fluctuations (RMSFs), the radius of gyration (Rg), the secondary structure according to the rules defined by the Kabsch and Sander DSSP program [42], and the solvent accessible surface area (SASA) over the course of the simulations. The evaluation of the structural drift was performed by measuring the RMSD of the Cα atoms with respect to their positions at time 0 (Figure 4B–F). The RMSD was calculated for the full-length protein (Figure 4B) and for the different regions in CD44s (Figure 4C,D). After an equilibration of 50 ns, the RMSD reached a plateau with a stable value of 24.3 ± 2.12 Å (Figure 4B). The RMSD of the 1UUH structure through the 500 ns trajectory was also calculated with respect to its corresponding initial minimized structure (Figure 4B). The modeled HABD (Figure 4C) adopted a stable conformation after 15 ns, with an RMSD of 1.7 ± 0.23 Å and presenting a smaller deviation with respect to the crystallographic HABD whose RMSD converged after 2.5 ns to around 2.5 ± 0.22 Å (Figure 4B). The transmembrane region was also quite stable and fluctuated around 2.1 ± 0.27 Å (Figure 4E). Figure 4 clearly indicates that high RMSD values of the whole protein were mainly due to the unstructured regions of CD44s (Figure 4D,F), meaning that most of the flexibility of the protein comes from the membrane–proximal stem region (170–268) and the intracellular domain (290–361), which, after 30 ns, reached the plateaus of 19.3 ± 1.38 and 20.5 ± 1.91 Å, respectively (Figure 4D,F).

The peaks observed in the RMSF plot revealed the regions of high flexibility within the CD44 structure. The profile of the HABD Cα-RMSFs curves for the residues was comparable in the corresponding MD trajectories (Figure 5A), indicating that these residues (20–169) share similar fluctuations with a mean value of 4.4 ± 0.92 and 0.8 ± 0.4 Å for AlphaFold2 and 1UUH, respectively. Limited movements were also observed for the transmembrane helix (mean RMSF of 6.7 ± 0.52 Å), whereas the N-terminal, the stem region, and the intracellular domain displayed significantly higher RMSF values (Figure 5A). We then examined the differences in the secondary structure element (SSE) composition (helices and strands) over frames for each amino acid residue. Figure 5B shows that the alpha-helical and beta-strand regions of the crystallographic HABD persisted throughout the simulation and that no differences in the secondary structure content were observed between the predicted modeled structure and 1UUH (Figure 5B,C). Of note, MD simulations showed the clear stability of the 269–289 helix and confirmed the ability of AlphaFold2 to predict the transmembrane helix with high confidence (Figure 2B). The Rg and the SASA gave a global account of the general tertiary structure of the protein. The plot of Rg versus time is presented in Figure 5D. The Rg trajectory pattern did not show significant trajectory oscillations, with an average value of 46.31 ± 1.76 Å. The curves of SASA (Figure 5E) indicated that the exposed areas (both hydrophobic and hydrophilic) for the AlphaFold2 model were stable during the entire simulations (25269.2 ± 1300.38 Å^2^). The plots of replicate analysis are available in Appendix A. The results of the replicates were in general agreement with the data obtained in the first MD.

Overall, MD simulations validated the models of CD44s generated using AlphaFold2. The analysis of trajectories highlighted the stability of the modeled HABD and TMD, as well as the conformational flexibility and structural dynamics of the unstructured regions that may allow CD44 to perform its unconventional activity, as has been observed in cancer [43,44,45] and immune response [16,46,47]. The physiologic function of the disordered region is possibly not univocal. First, it is most likely involved in the interaction of CD44 with CD49 [48] which represents a co-receptor involved in the homing of bone-marrow-derived cells into the CNS during experimental multiple sclerosis. Second, all variant isoforms of CD44 differ in this region of the molecule. Several laboratories have shown that the expression of some variants is associated with the metastatic potential of human cancer cells [11,45,49]. We recently showed that the expression of mouse CD44 was specifically associated with the modification of the trafficking properties of T cells by changing the effect of its interaction with hyaluronic acid on cell motility and possibly other constituents of the extracellular matrix such as osteopontin [16]. Thus, it appears that this disordered region modulates the intracellular signals that are elicited by the binding of matrix components by the constant region.

### 3.3. AlphaFold2 Models of CD44v3-10 and CD44v7-10 Isoforms

All splicing isoforms of CD44 maintain the same ligand-binding and transmembrane/intra-cytoplasmic regions, while they vary in the intermediate disordered region. To understand whether the modification of the extra variable domains can lead to a better prediction of the structural order of CD44, we focused on two variants of particular interest: CD44v3-10 and CD44v7-10. CD44v3-10 is related to the metastatic potential of cancer cells [50] and the proliferation of endometrial stromal cells [51] and is involved in the infiltration of articular synovia in various autoimmune diseases [52]. CD44v7-10 is highly expressed by cells in the cerebrospinal fluid of MS patients selectively during the active phases of the disease, as we have recently reported [16]. Furthermore, T cells are licensed to upregulate the production of this isoform only in MS patients and only at the acute presentation of the disease.

In agreement with the model of CD44s, AlphaFold2 provided high-confidence predictions of the HABD (20–169) and transmembrane helix (269–289) of CD44v7-10 and CD44v3-10 (Figure 6). The insertion of variant exons into the CD44 stem region (170–399 and 170–606 in CD44v7-10 and CD44v3-10, respectively) lengthened the size of the disordered region in the extracellular domain (Figure 2B and Figure 6 for comparison).

## 4. Conclusions

The structural characterization of CD44 is an important task, because this receptor is involved in many physiological and pathological processes, including inflammation, immune response, and cancer progression [6,19,43,53]. Here, we present the first structural model of whole-length CD44s using deep learning techniques, namely D-I-TASSER, AlphaFold2, and RoseTTAFold. AlphaFold2 was able to predict the 3D coordinates of the folded hyaluronan-binding domain (HABD) with a root mean square deviation (RMSD) of 0.8Å compared to the experimental structure (PDB entry 1UUH). In particular, the HABD model structure generated by AlphaFold2 exhibited equivalent stereochemical parameters of the crystallographic 1UUH structure. AlphaFold2 was also able to predict, with high confidence, the transmembrane alpha helix spanning residues 269–289 which remained stable, with HABD, until the end of the MD simulations carried out to assess the stability of predicted model. This study confirms the ability of AlphaFold2 to predict protein structures with very high accuracy and identify largely unstructured regions. AlphaFold2 analysis on CD44v3-10 and CD44v7-10 isoforms clearly indicated that the insertion of variant exons increases disorder in the stem region, whose crucial role in substrate recognition has been demonstrated. Our data also represent the basis for future studies aimed at characterizing CD44 isoforms and identifying new potential therapeutic agents targeting CD44.

## Figures and Tables

**Figure 1 biomolecules-13-01047-f001:**
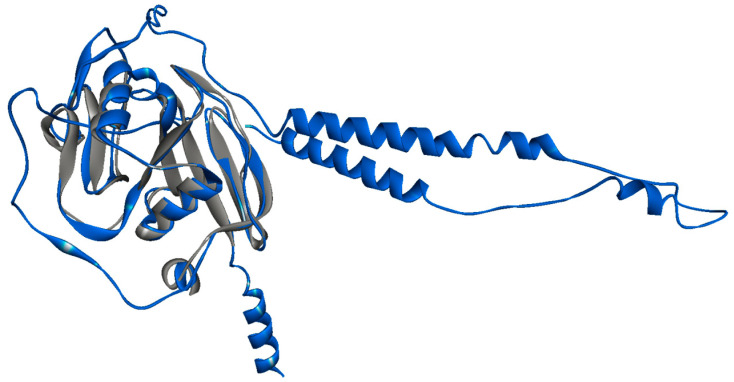
Structural model of CD44s generated via D-I-TASSER (blue). The crystallographic HABD (pdb entry 1UUH) is superimposed (gray).

**Figure 2 biomolecules-13-01047-f002:**
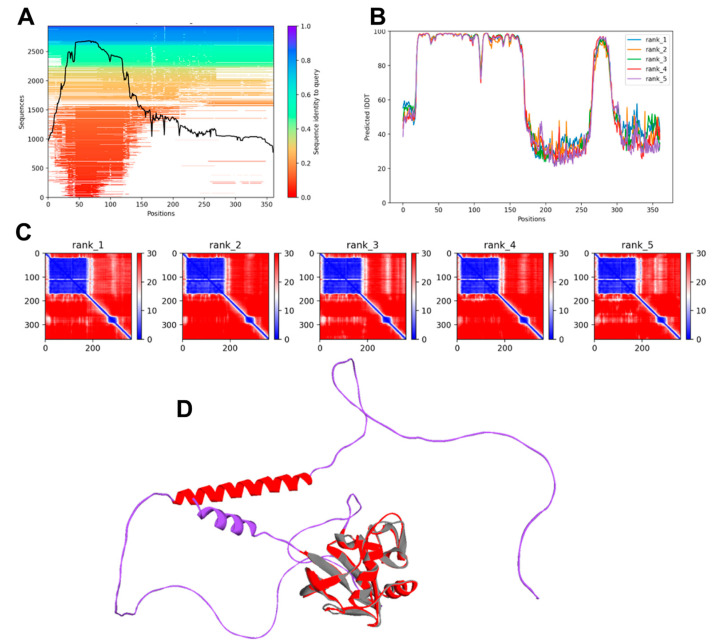
The output of AlphaFold2. (**A**) Multiple sequence alignment (MSA) for the prediction of the CD44 structure. The number of sampled sequences is plotted on the *y* axis against the amino acid position on the *x* axis. The sequence identity to the queried sequence is indicated by the bar on the right, color coded from red (low identity) to blue (high identity). Coverage is shown by the black line. (**B**) Predicted local distance difference test (LDDT) score vs. position for the five models generated via AlphaFold2. (**C**) Prediction alignment error (PAE) score for the five models generated via AlphaFold2. The axes indicate the position of the amino acids. Reliability of pairwise relative positions of amino acids is color coded from blue (0 Å) to red (30 Å), as shown in the right bar. (**D**) The whole-length-predicted structural model of CD44s: good- and low-confidence regions are shown in red and purple, respectively. The superimposed crystallographic structure of HABD (pdb entry 1UUH) is shown in gray.

**Figure 3 biomolecules-13-01047-f003:**
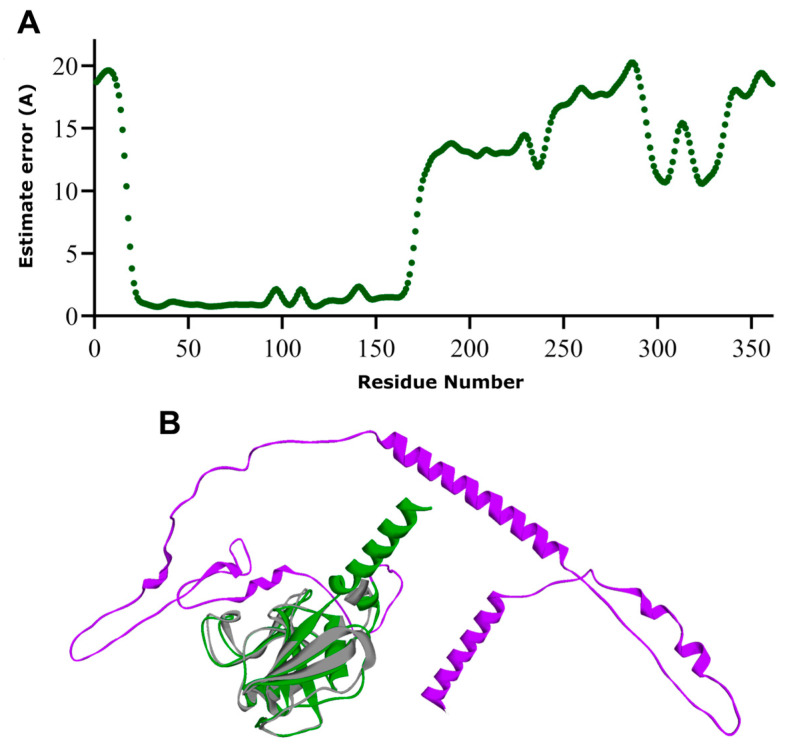
The output of RoseTTAFold. (**A**) Local quality of the model expressed in per-residue error estimate. (**B**) The whole-length-predicted structural model of CD44s colored by local model quality: high-quality and low-quality in green and purple, respectively. The superimposed crystallographic structure of HABD (pdb entry 1UUH) is shown in gray.

**Figure 4 biomolecules-13-01047-f004:**
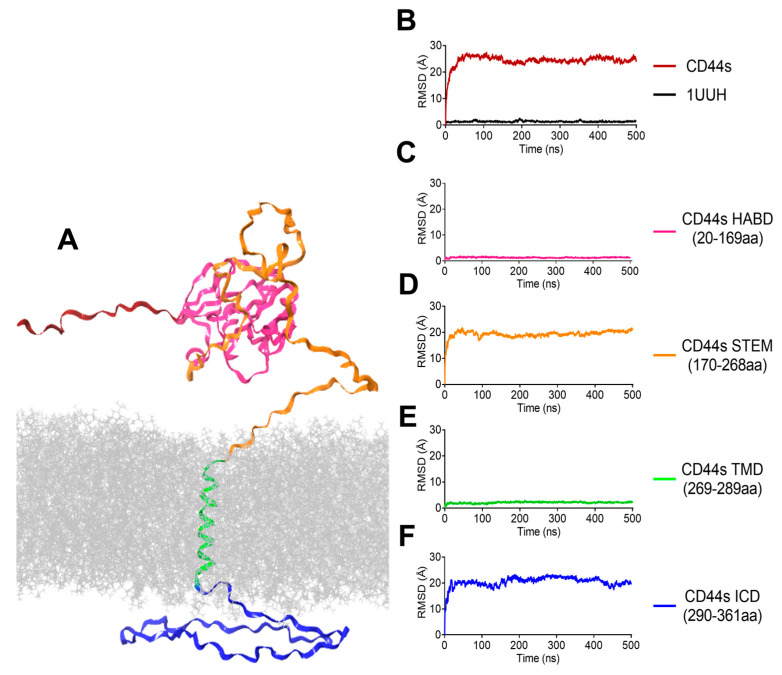
Root mean square deviation (RMSD) for CD44 simulation. (**A**) The input structure for MD simulation: CD44s in the membrane. HABD is colored red, the stem region orange, TMD green, and ICD cyan. Time series of the Cα atoms’ RMSD from the starting structure are shown for (**B**) the full protein and 1UUH, (**C**) HABD, (**D**) stem region, (**E**) TMD, and (**F**) ICD.

**Figure 5 biomolecules-13-01047-f005:**
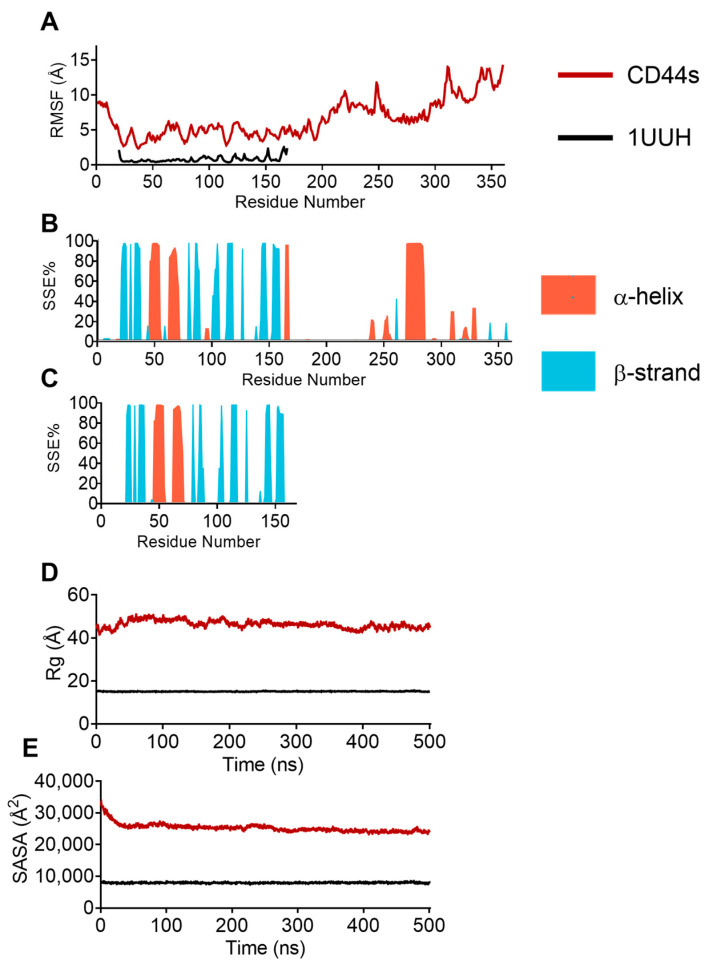
Analysis of MD trajectories. (**A**) Residue-based Cα-RMSF relative to the starting structure. (**B**) Distribution of secondary structure elements (SSEs) via residue index throughout the structure of CD44s. (**C**) Distribution of secondary structure elements’ (SSEs’) distribution via residue index throughout 1UUH. (**D**) Time series of the radius of gyration of Cα atoms. (**E**) Time evolution of the solvent accessible surface area (SASA).

**Figure 6 biomolecules-13-01047-f006:**
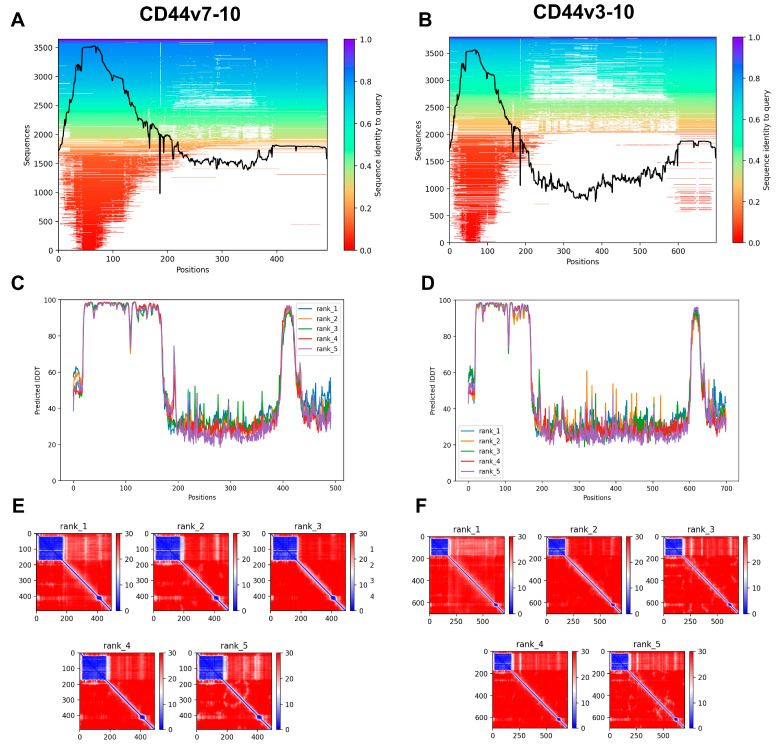
The output of AlphaFold2 for the CD44v7-10 and CD44v3-10. (**A**–**C**) Multiple sequence alignment (MSA). The number of sampled sequences is plotted on the y axis against the amino acid position on the *x* axis. The sequence identity to the queried sequence is indicated by the bar on the right, color coded from red (low identity) to blue (high identity). Coverage is shown by the black line. (**D**–**F**) Predicted local distance difference test (LDDT) score vs. position for the five models generated via AlphaFold2. Prediction aligned error (PAE) score for the five models generated via AlphaFold2. The axes indicate the position of the amino acids. The reliability of the pairwise relative positions of amino acids is color coded from blue (0 Å) to red (30 Å), as shown in the right bar.

**Table 1 biomolecules-13-01047-t001:** Details of the starting structures for MD simulations.

Structure	No. of Atoms	No. of POPC Molecules	No. of Water Molecules	No. of Cl^−^ Ions	No. of Na^+^ Ions	Box Size (Å)
AlphaFold2	229,696	472	53,575	149	162	102 × 83 × 148
1UUH	19,713	-	5809	16	19	57 × 68 × 55

**Table 2 biomolecules-13-01047-t002:** Evaluation of HABD models by using PROCHECK, TM and Molprobity scores.

Ramachandran Plot Statistics (%)					TM Score ^a^	MolProbity Score ^b^
	Most Favored	Allowed	Generously Allowed	Disallowed		
D-I-TASSER	55.1%	33.3%	8.0%	3.6%	0.87	3.24
AlphaFold2	86.3%	13.7%	0.0%	0.0%	0.91	1.57
RoseTTAFold	82.4%	14.5%	1.5%	1.5%	0.87	1.48
1UUH	85.9%	14.1%	0.0%	0.0%	1.0	2.39

^a^ TM scores range from 0 to 1, where 1 is the highest accuracy. ^b^ A low MolProbity score indicates that a model is more physically favorable.

## Data Availability

The data are available upon reasonable request.

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
