# Peer review of "Prediction of CD44 Structure by Deep Learning-Based Protein Modeling"

_biomolecules, 2023, doi:10.3390/biom13071047_

Round 1

Reviewer 1 Report (Previous Reviewer 1)

The paper addressed most of my questions, but there are missing details that could make the paper stronger.

1.    The prediction of CD44s protein structure is meaningless, because the HABD region has experimental structure, and all others are coil except one short transmembrane helix region. 

2.     The authors run AlphaFold2 and RosettaFold for CD44 based on online servers, so both should use native (1UUH) information in modeling. So it does not make sense to use model RMSD to 1UUH as a measure to select RosettaFold model as the best model, except first build the model without native. 

3.     In line 394, it should be RoseTTAFold instead of AlphaFold2. 

4.     What is HAB domain in line 364 and 487.

There are some typos. Please further do proofreading by a native English speaker to improve grammar and typo.

Author Response

Point 1.    The prediction of the structure of CD44 protein is meaningless, because the HABD region has an experimental structure, and all other regions are coil except one short transmembrane helix region.

Response to Point 1. This manuscript evaluates the ability of AlphaFold2, in comparison with other deep learning approaches, namely RoseTTAFold and D-I-TASSER, in predicting the structural characteristics of CD44. Variations in the ratio between CD44 and CD44 isoforms play an important role in several pathological processes, as widely described in the manuscript.  Our study, with the support of AlphaFold2, demonstrates that insertion of variant exons increases disorder in the stem region, whose crucial role in substrate recognition is known.  For the above reasons we believe that our manuscript largely fits with the objective of this special issue. 

Point 2.     The authors run AlphaFold2 and RosettaFold for CD44 based on online servers, so both should use native (1UUH) information in modeling. So, it does not make sense to use model RMSD to 1UUH as a measure to select RosettaFold model as the best model, except first build the model without native.

Response to Point 2. Actually we did not select RosettaFold model as the best model. In fact, only the AlphaFold2 model was submitted to MD simulations and reported along with a thoroughly discussion of its dynamic behaviour in lipid bilayer. As written in the manuscript and clearly summarized in the abstract and conclusions, AlphaFold2 (and not RosettaFold) performs better on the basis of several analyses, which, in addition to the informative RMSD to 1UUH, include stereochemical parameters and the high confidence prediction of the transmembrane helix.

Point 3.     In line 394, it should be RoseTTAFold instead of AlphaFold2.

Response to Point 3. We would need other details to identify the sentence since, in the downloaded revised version, 394 refers to the Conflict of Interest statement.

Point 4.     What is the HAB domain on lines 364 and 487.

Response to Point 4. “HAB domain” is HABD. We replaced “HAB domain” with HABD at lines 205 and 283 of the revised manuscript.

Typos and English were checked

Reviewer 2 Report (Previous Reviewer 2)

In this revised version, the authors have answered all my comments. In addition, they have greatly improved the paper, including a better algorithm for three-dimensional structure predictions. Also, a more realistic model of the receptor bound to a membrane has been incorporated. The paper represents a sound contribution and merits publication in its present form.

Author Response

We really thank the reviewer for his comments.

Round 2

Reviewer 1 Report (Previous Reviewer 1)

All questions are addressed. 

This manuscript is a resubmission of an earlier submission. The following is a list of the peer review reports and author responses from that submission.

Round 1

Reviewer 1 Report

The paper employed the recently developed deep learning-based AlphaFold2 and RoseTTAFold tools to predict the whole length CD44s. Both approaches correctly predicted the HABD, with AlphaFold2 outperforming RoseTTAFold in the structural comparison with the experimental HABD structure. 

Unfortunately, this paper is lack of novelty since it only ran some recently developed tools for only one protein. There are several major problems and some fundamental concerns with the experimental design and the writing. This means the strong conclusions put forward by this manuscript are not warranted and I cannot approve the manuscript in this form. Please see the following comments.

Major:

1.     The prediction of CD44s protein structure is meaningless, because the HABD region has experimental structure,  and all others are coil except one short transmembrane helix region. 

2.     Since the authors want to investigate the CD44, they should do prediction and analysis for all isoforms in Table S1 instead of only P16070-12. 

3.     In line 71-72, the authors said “I-TASSER ranked as the 71 No 1 server for protein structure prediction in the CASP7-CASP14 experiments.” Note that I-TASSER participated CASP7-CASP11. C-I-TASSER, which developed based on I-TASSER, participated CASP12-CASP13. Similarly, D-I-TASSER participated in CAS14-CASP15. I-TASSER is purely template-based method, while C-I-TASSER and D-I-TASSER have deep learning module. So simply say I-TASSER ranked as best server in CASP7-CASP14 is not correct, please correct it. 

4.     The authors can try to use the D-I-TASSER, which also includes deep learning modules, to predict structure model for CD44 instead of I-TASSER. 

5.     The authors run AlphaFold2 and RosettaFold for CD44 based on online servers, so both should use native (1UUH) information in modeling. So it does not make sense to use model RMSD to 1UUH as a measure to select RosettaFold model as the best model, except first build the model without native. 

6.     In line 138-139, it should be predicted TM-score here. Please correct it.  

7.     In line 221, it should be RoseTTAFold instead of AlphaFold2. 

8.     In table 2, please also report TM-score and molprobity score of the predicted structure models. 

9.     Please clarify which MD replicate is shown in Figure3, and the difference the figure 3 and figure S5-S8 

Minor:

1.     In line 65, missing “d” for “RoseTTAFold”. Please check the typos all over the manuscript and supplementary materials. 

2.     In figure 1A-1C, the texts in axis are too small to read. 

3.     In figure 3, the texts in axis are not in the same size, which makes the figure messy and ugly. And the helix legend in figure 3C is mixed with some blue points.

4.     In figure S1, Table S2 and Table S3, the authors use “favoured”, while in the main manuscript, use “favored”. Please use a consistent form.

5.     In figure S5 and S7, what the RMSD2 and RMSD3 meaning? And why don’t have RMSD1? Please clarify the details in figure legends. 

6.     In figure S6 and S8, what the color meaning? And why have two plots for each of methods? Please clarify the details in figure legends. 

7.     In figure S7, the figure is overlapped with the legends. 

8.     The figure S2, S6, and S8 are in low resolution and difficult to read texts in figure.

Reviewer 3 Report

In this manuscript, Camponeschi et al. predict CD44s isoform structure using AlphaFold2, RosettaFold and I-TASSER methods and run MD simulations on generated models. Although the modelling of such an important cell surface protein as CD44 is very much to be welcomed, the work presented contains very large errors and misinterpretations of the data. Furthermore, after reading the manuscript, the reader did not know more than CD44s contains a folded hyaluronan-binding domain and transmembrane domain, while the rest of the molecule is disordered. However, this structural description is obviously seen in the deposited structure of another isoform of CD44 (see https://alphafold.ebi.ac.uk/entry/P16070), so the manuscript has no additional scientific value.

Major drawbacks:

- CD44 is a transmembrane protein with a single transmembrane segment. Performing MD simulation and stability analysis on the whole structure in a water-filled box is a major scientific error. It should be embedded in a double lipid layer using e.g. CHARM (https://charmm-gui.org), or the transmembrane segment should be cut off before simulation.

- It has been shown that AlphaFold2 based predictions of protein disorder outperform dedicated tools (see e.g. PMID: 36344848), so there is no point in using such a tool (PONDR) after AF2 prediction. Moreover, PONDR (and almost all other IDR prediction methods) were developed to predict dirodered regions in globular proteins and their accuracy on TMPs is rather limited.

- CD44 has an N-terminal cleavable signal sequence, so models that include the signal part are obviously erroneous.